# AIQS-DB: Revolutionizing the Simultaneous Analysis of Organic Compounds

**Quang Minh Bui \*, Huynh Nhat Minh Nguyen, Van Nhan Le** **, Thanh Thao Nguyen, Ngoc Minh Truong, Ngoc Tung Nguyen** **, Quang Huong Le** **and Quang Trung Nguyen \***

Center for Research and Technology Transfer, Vietnam Academy of Science and Technology, 18 Hoang Quoc Viet, Cau Giay, Hanoi 11353, Vietnam; joshep.minhnguyen2812@gmail.com (H.N.M.N.); levannhan.na@gmail.com (V.N.L.); thao7980@gmail.com (T.T.N.); anphuminh1011@gmail.com (N.M.T.); tungnguyen.vast@gmail.com (N.T.N.); lehuong3795@gmail.com (Q.H.L.)
\* Correspondence: bui_quang_minh@yahoo.com (Q.M.B.); nqtrung79@gmail.com (Q.T.N.); Tel.: +84-98-517-3286 (Q.M.B.)

**Abstract:** This paper reports a database, namely, the Automated Identification and Quantification Database System (AIQS-DB), which consists of three components, including retention times, mass data, and calibration curves, without the requirement to analyze standard substances. The AIQS-DB that are pre-registered in the database are used as the replacement for the process of measuring chemical standards. Both the target and unknown substances in the real samples were determined by the same conditions of GC-MS as those used for the initial database register in the AIQS-DB system. The article provides a comprehensive overview of the wide-ranging applications of AIQS-DB in various fields and highlights its usefulness as a tool for the simultaneous analysis of organic compounds in different matrixes such as water, soil, sediment and air, etc. It could be considered as the basis in further applications of the AIQS-DB method in determining organic compounds in other fields, specifically biology, food, agriculture, medicine, etc., allowing assessment and reflection on the quality and status of the studied products quickly and cost-effectively.

**Keywords:** AIQS-DB; GC/MS; LC/MS; simultaneous analysis; organic compounds

## 1. Introduction

Typically, a calibration curve is utilized to determine chemical compositions in actual samples analysis in the GC-MS system. However, a calibration curve being generated requires cost, time and effort as well as the disposal of substances and solvent. In addition, the calibration curve method cannot be applied to measure more than about 100 substances simultaneously.

The Automated Identification and Quantification Database system (AIQS-DB) is a database comprising three components, including retention times, mass information, and calibration curves, without requirements to analyze standard substances. The data that are pre-registered in the database are used as the replacement for the process of measuring chemical standards. Theoretically, all substances were measured with the same GC-MS conditions as those utilized for initial registration in the database, which can be determined by AIQS-DB system. In addition, not only target substances but also unknown substances in the real samples were searched for using the AIQS-DB. In 2005, Professor Kadodami Kiwao, a distinguished academic at Kitakyushu University, introduced groundbreaking research on AIQS-DB [1], a sophisticated technology that leverages GC/MS analytical instruments to detect hundreds of organic compounds in a single sweep. The development of AIQS-DB marked a significant milestone in the field of analytical chemistry, enabling researchers to identify an unprecedented number of organic compounds with greater efficiency and accuracy than ever before.

At its inception, AIQS-DB boasted a remarkable capacity to identify 672 compounds simultaneously, including 332 types of pesticides, 160 varieties of polycyclic aromatic hydrocarbons (PAHs) and polychlorinated biphenyls (PCBs) composed of carbon and hydrogen, 81 phenols comprised of carbon, hydrogen, and oxygen, and 99 compounds containing nitrogen, sulfur, and phosphorus [1]. However, it was not until 2010 that AIQS-DB was refined and gradually adopted for analyzing diverse sample matrices [2].

Since its inception, AIQS-DB has been widely utilized in numerous studies to concurrently analyze organic compounds in various sample matrices, such as river water samples [3–6], groundwater samples [7–9], sediment samples [10–12], wastewater samples [13–15], and dust samples [16–18]. Moreover, AIQS-DB has been adapted for use with other analytical instruments, including LC/MS/MS and LC-TOF/MS, making it an expedient, comprehensive, cost-effective, and scalable approach for identifying organic micro-pollutants [6]. The ability of AIQS-DB to analyze up to 970 compounds simultaneously in GC/MS and 508 compounds in LC-QTOF-MS or LC/MS makes it an ideal tool for non-target analysis across a broad range of sample matrices and applications.

This article provides a comprehensive overview of the wide-ranging applications of AIQS-DB in various fields and highlights its usefulness as a tool for the simultaneous analysis of organic compounds. It aims to promote the use of AIQS-DB, a highly useful product of Professor Kadokami's research, and encourages further research on its applications in other sample matrices such as food, agricultural products, and plants.

## 2. AIQS-DB Application

A gas chromatography mass spectrometry (GC/MS) coupled with the Automated Identification and Quantification System (AIQS) was created and developed by Kadokami et al., 2005 [1], to simultaneously identify and quantify almost 1000 semi-volatile organic compounds in various matrixes. The AIQS database contains the target compound calibration curves, which were created by utilizing the conventional internal standard-based method, and the AIQS was applied to identify and quantify target compounds in environmental samples. Therefore, analytical standards for target compounds that were registered in the AIQS were not used to make calibration curves in the analysis of samples when using the AIQS; hence, the instrument conditions and tuning of the GC/MS system must be the same as those of the GC/MS system used to create the entries in the AIQS database to obtain accurate results [7].

AIQS-DB, when used on a GC instrument, is commonly referred to as AIQS-GC, whereas its application on an LC system is known as AIQS-LC [19,20]. Recently, AIQS-DB has been adopted for Comprehensive Target Analysis, leading to the development of various shorthand terminologies, including Comprehensive Target Analysis with an Automated Identification and Quantification System (CTA-AIQS) [21–23], target screening analysis (TSA-AIQS) [23], and the comprehensive screening method (CSM-AIQS) [24], which are applicable to both GC and LC modalities. In light of its rapid deployment for environmental micro-pollutant screening in emergency scenarios, AIQS-DB has been abbreviated as REPE [23]. Numerous organic compounds in various matrixes were measured under GC-MS and LC-MS coupled with the AIQS-DB system, and are presented in Table 1.

**Table 1.** The research used AIQS-DB to determine organic compounds in various matrixes.

| Year | Instrument | Compound Number | Matrix | Main Objective | Ref. |
|---|---|---|---|---|---|
| 2023 | GC/MS | 58/949 | Wastewater, treated water | Assessment of the influencing factors of ozonation performance in removing CoC in a wastewater discharge | [25] |
| 2023 | LC-QTOF-MS | 125/484 | River water | Analytical method development for LC-QTOF-MS | [6] |
| 2023 | GC/MS; LC/QTOF-MS | 144/969; 69/421 | River water | Development of AIQS-DB for passive sampling as CC and POCIS | [3] |
| 2022 | GC/MS | 288/N.a | PM 2.5 | Comprehensive analysis | [16] |
| 2022 | LC-QTOF-MS | 57/508 | Indoor dust | Comprehensive analysis and health risk assessment | [26] |
| 2022 | GC/MS | 97/886 | Indoor air and dust samples | Comprehensive analysis and health risk assessment | [18] |
| 2022 | GC/MS; LC/MS | 133/969 | Dust samples | Comprehensive analysis and health risk assessment | [17] |
| 2022 | GC/MS | 32/886 | Wastewater treatment effluent | Profiling of organic pollutants | [27] |
| 2022 | GC/MS | 109/~1000 | Flood sediment or soil samples | Risk assessment | [20] |
| 2022 | LC-QTOF-MS | 20/296 pesticides | Surface water samples in agriculture area | Comprehensive and agrochemical analysis | [22] |
| 2021 | LC-QTOF-MS | 22/187 | Particle samples | Risk assessment | [28] |
| 2021 | GC/MS; LC/QTOF-MS | 78/970 2/501 | Sediment | Comprehensive analysis | [21] |
| 2021 | LC-QTOF-MS | 19/107 | Particle samples | Comprehensive analysis and risk assessment | [29] |
| 2021 | GC/MS; LC/QTOF-MS | 474/970 | Wastewater eluted by fire extinguishing activities and river water | Comparison with GC-QTofMS | [19] |
| 2021 | GC/MS | 136/948 | River water | Analytical method development | [18] |
| 2021 | GC/MS; LC/QTOF-MS | 131/948 311 | River water | Screening and ecological risk | [22] |
| 2020 | LC-Q/TOF-MS | 85/484 | River water | Comprehensive survey | [30] |
| 2019 | GC/MS | 195/942 | Indoor dust | Comprehensive analysis | [31] |
| 2019 | GC/MS | 118/970 | Particle samples | Target screening analysis | [32] |
| 2019 | GC/MS | 167/942 | Passive air sampling | Comprehensive analysis and health risk assessment | [33] |
| 2019 | GC/MS | 105/942 | Road dust samples | Comprehensive analysis and health risk assessment | [34] |
| 2019 | LC-Q/TOF-MS | 201/484 | Wastewater of a sewage treatment plant | Comprehensive target analysis | [15] |
| 2019 | GC/MS | 63/937 | Tsunami sediment samples | Comprehensive screening and risk assessment | [23] |
| 2018 | GC/MS | 127/940 | Surface river water | Comprehensive screening and risk assessment | [4] |
| 2018 | GC/MS; LC/QTOF-MS | 165/1153 | River water | Comprehensive screening and risk assessment | [5] |
| 2018 | GC/MS | 196/943 | Municipal wastewater | Comprehensive analysis | [14] |

**Table 1.** *Cont.*

| Year | Instrument | Compound Number | Matrix | Main Objective | Ref. |
|------|-----------|-----------------|--------|----------------|------|
| 2018 | GC-MS<br>LC-MS | 109/1250 | Wastewater | Assessment of the efficiency of wastewater treatment system | [13] |
| 2018 | GC-MS | Used for only PAH and OCP | Soils and sediments | Analytical method development | [35] |
| 2017 | GC-MS | 277/940 | Floodwater | Comprehensive analysis | [36] |
| 2016 | GC-MS<br>LC-MS | 78/1300 | Groundwater | Comprehensive screening | [8] |
| 2016 | GC-MS<br>LC/QTOF-MS<br>LC-MS | 80/1300 | Groundwater | Comprehensive screening | [9] |
| 2015 | GC-MS<br>LC-MS | 227/1300 | River water | Water monitoring | [37] |
| 2015 | GC-MS | 74/940 | Groundwater | Comprehensive screening | [7] |
| 2014 | GC-MS | 185/940 | River sediment | Comprehensive screening | [38] |
| 2014 | GC-MS | 195/940 | Sediment | Comprehensive screening | [12] |
| 2014 | GC-MS | 95/940 | River water | Comprehensive screening | [39] |
| 2013 | GC-MS | 184/888 | Sediment | Comprehensive screening | [10] |
| 2012 | GC-MS | 914 | Sediment | Analytical method development | [11] |
| 2011 | GC-MS | 114 | N.a. | Verification of analytical method | [40] |
| 2010 | GC-MS | 95/940 | River water | Comprehensive screening | [2] |
| 2009 | GC-MS | 188/882 | River water | Comprehensive screening | [41] |
| 2005 | GC-MS | 13/672<br>56/672<br>150/672<br>150/672 | River water<br>Soil<br>Spinach<br>Orange | Analytical method development | [1] |

N.a.: Not available.

Among the studies conducted, four studies have focused on the development of analytical methods, primarily using GC/MS instruments. The first study, published in 2005, analyzed only 672 compounds, and showed the potential for applications in environmental matrices such as river water, soil, and food samples such as spinach and oranges [1]. In 2011, Terumi Miyazaki et al. tested the accuracy of the AIQS-DB method [40], focusing on 114 compound groups of organochlorine pesticides and PAHs. The results showed a high level of accuracy. In 2012, Kadokami published another study that expanded the AIQS-DB parameters to analyze up to 914 compounds and applied it to sediment analysis [11]. The maximum number of compounds that can be analyzed using AIQS-DB is 970 [19,21,32]. The application of AIQS-DB to LC/MS and LC-TOF-MS was first published in 2016 by Xuehua et al., with a range of over 300 compounds [9]. LC/MS and LC-TOF-MS are less frequently used than GC/MS, but up to 508 compounds have been constructed using these methods [26]. In total, both methods can analyze 1478 compounds.

The number of compounds detected depends on the sample matrix. For example, in wastewater matrices analyzed by GC/MS in Japan, Ryo Omagari was able to detect the presence of 474 compounds in the influent wastewater soluble fraction [19]. In dust samples collected in Vietnam, Le Quang Huong et al. identified 288 compounds [16], and Trinh Thu Ha et al. identified 277 compounds in flood water samples [36]. Almost 200 compounds can be detected in sediment samples [10,12,38]. For samples analyzed by LC, around 201/484 compounds can be detected in wastewater samples [15]. AIQS-DB on LC/MS and LC-TOF-MS is often able to identify fewer compounds, which may be why studies using AIQS-LC are less common than those using AIQS-GC. The number of identified compounds demonstrates that AIQS-DB is very useful for the simultaneous determination of pollutants in the environment.

In the nearly 20 years since GS. Kadokami's study was published, AIQS-DB has only been applied in studies on environmental samples collected in a few countries, such as Australia [3,13], China [2,8,14,27], Japan [10,12,39], Malaysia [17], Serbia [4], and Vietnam [5,7,28,31–33,36,38]. The application of AIQS-DB in environmental studies should be expanded to more countries.

### 3. Summary of AIQS Mechanisms and Instrumental Conditions

The AIQS-DB method, described by Ryo Omagari et al. in their research, utilizes retention times, mass spectra, and internal standard calibration curves stored in a database to identify and quantify chemical compounds [1,19]. To achieve accurate results, the GC-MS instrument must be adjusted to specific conditions. The AIQS-GC and AIQS-LC methods were validated through the analysis of procedural blanks, duplicate samples, and certified reference materials. The beauty of the AIQS theory lies in the fact that if the measurement equipment conditions remain constant, the retention times and calibration curves of chemicals remain unchanged, thereby eliminating the need for standard chemical preparation. During AIQS analysis, the target ion's peak in a sample is located, and the target is identified by assessing the similarity between actual and predicted values using the extracted spectrum. Based on the accuracy of identification, the target is then given a rating of 1–5 stars. The AIQS-DB method holds great promise for revolutionizing the analysis of organic compounds.

Studies using the AIQS-DB method were all conducted using the GC or LC methods developed by GS. Kadokami. Therefore, publications only present quality assurance and quality control procedures. Analytical conditions for GC/MS and LC-TOF-MS instruments are briefly presented in the Table 2, while detailed equipment conditions are discussed below.

**Table 2.** Summary of GC-MS and LC-TOF-MS conditions.

| Item | GC-MS Specification [1,21,22] | LC-TOF-MS Specification [6,21] |
|---|---|---|
| *Company* | Shimadzu | Sciex, Agilent, Shimadzu |
| *Column* | DB-5 MS (30 × 0.25 × 0.25) | ODS (2.1 × 150 × 3) at 40 °C |
| *Temperature program/gradient* | 2 min at 40 °C, 8 °C/min to 310 °C, 5 min at 310 °C; | A95:B5 (0′)–A5:B95 (30′–50′)<br>Flow Rate: 0.3 mL/min |
| *Injection:* | 250 °C/*splitless* | |
| *Transfer line* | 300 °C | |
| *Ion source* | 200 °C | |
| *Carrier gas/mobile phase* | He | $H_2O$ (A): $CH_3OH$ (B) + 5 mmol $CH_3COONH_4$ |
| *Linear velocity* | 40 cm/s, constant | |
| *Ionization* | EI | ESI-Positive at 3500 V |
| *Mode* | SIM/SCAN (400–600 aum) | SCAN ($m/z$ 50–1000) |

### 4. Prospects

The advantage of AIQS-DB is that it can simultaneously analyze up to 970 compounds on GC/MS equipment and 508 compounds on LC-QTOF-MS or LC/MS equipment. When combining the analysis results from AIQS-DB with multivariate statistics such as PCA or LDA, the geographical origin of the sample can be traced. Origin tracing studies have been being developed for over 15 years. The types of samples used in origin tracing studies include food samples, agricultural products, rice samples, wine samples, meat samples, etc. The analysis techniques used in origin tracing are also diverse, such as GC/MS, LC/MS, ICP/MS, Isotope, FTIR, etc. The methods can be accurate concentration analysis, semi-quantitative analysis, or even just intensity signal analysis. All are methods that analyze multiple elements and compounds. Therefore, the AIQS-DB method is very suitable for applications in origin-tracing studies.

AIQS-DB is a highly promising technology that has the potential to revolutionize the way we determine food's geographical origin traceability and identify bioactive compounds in plant medicine. The method combines retention times, mass spectra, and internal

standard calibration curves registered in a database to identify and quantify chemical substances with high accuracy and reliability.

One of the major advantages of AIQS-DB is that it requires minimal preparation of standard chemicals, as the retention times and calibration curves remain constant if the measurement conditions are consistent. This saves time and resources, while ensuring that results are consistent and reproducible.

In terms of food geographical origin traceability, AIQS-DB can be used to accurately identify the geographic origin of food products based on the chemical composition of the sample. This can be especially important for products with geographical indications, such as wines, cheeses, and meats, where consumers rely on the origin to determine the quality and authenticity of the product. With AIQS-DB, we can ensure that the geographical origin of a product is accurately determined, which can help to prevent fraud and protect the reputation of the product.

AIQS-DB is also highly useful in the field of plant medicine, where it can be used to identify bioactive compounds that have therapeutic properties. Traditional methods of identifying bioactive compounds can be time-consuming and require a significant amount of resources. With AIQS-DB, we can quickly and accurately identify bioactive compounds, allowing us to develop more effective treatments and therapies for various medical conditions.

Overall, the potential applications of AIQS-DB are numerous and diverse. By providing accurate and reliable results with minimal preparation of standard chemicals, this technology has the potential to transform the way we determine food geographical origin traceability and identify bioactive compounds in plant medicine, leading to improved quality control and better health outcomes.

## 5. Conclusions

The paper shows the wide-ranging applications of AIQS-DB in determining organic compounds in different environmental matrixes, which has great potential to be applied in various fields due to the AIQS-DB being able to simultaneously analyze up to 1500 compounds. The AIQS-DB can be considered as the largest chemical analysis database system in the world. However, until now, there have been few studies (around 40 research works) applying the AIQS-DB. Therefore, this article is eager for research applying AIQD-DB to be further developed and applied for various purposes, such as in food analysis and source tracing. Moreover, developing a suitable dataset could potentially reduce the time and cost for discovering new bioactive compounds in plant medicine, as well as reducing the chemical waste from studies discharging to the environment.

**Author Contributions:** Conceptualization, Q.T.N. and Q.M.B.; resources, H.N.M.N., N.T.N. and Q.H.L.; data curation, T.T.N. and N.M.T.; writing—original draft preparation, V.N.L. and Q.T.N.; writing—review and editing, V.N.L. and Q.T.N.; visualization, Q.M.B.; supervision, Q.T.N.; project administration, Q.M.B.; funding acquisition, Q.M.B. All authors have read and agreed to the published version of the manuscript.

**Funding:** This research was funded by the Vietnam Academy of Science and Technology under the project with fund number code NCXS 01.02/23-25.

**Institutional Review Board Statement:** Not applicable.

**Informed Consent Statement:** Not applicable.

**Data Availability Statement:** Not applicable.

**Conflicts of Interest:** The authors declare that there are no conflict of interest regarding the publication of this paper.

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
