# Peer review of "AIQS-DB: Revolutionizing the Simultaneous Analysis of Organic Compounds"

_applsci, doi:10.3390/app13148031_

Round 1

Reviewer 1 Report

The authors try to show the benefits of using this tool for the detection of organic compounds in environmental samples quickly and efficiently, however the document presented does not have the characteristics or the depth that a review article must have. For example, the theoretical principles of the coupling of said tool to a chromatographic system are not explained, likewise the authors give examples of the application of this tool in Table 1; however, instrumental details of each of the studies are not given, nor are the compounds identified at their different concentration levels. In summary, what the authors present is not enough even for a short communication.

Author Response

The response is in attach file.

Reviewer 2 Report

THE BIBLIOGRAPHY MUST BE MODIFIED ACCORDING TO THE INSTRUCTIONS

Author Response

The response is in attach file.

Reviewer 3 Report

Comments: The article entitled “AIQS-DB: Revolutionizing Simultaneous Analysis of Organic 2 Compounds” is overall a good scientific effort that has been made by Bui Quang Minh et al. The background and purpose of this study is interesting and the research article is properly handled by the authors. It could help the fast track research for the scientific community and economically discover the new bioactive compounds from medicinal plants. To sum up, I believe that this short communication fulfills the quality to be published in Journal of Applied Sciences and can be of interest for its readership.

Recommendation: Accept with major revision

Regarding formatting and style: The authors need to follow the general format of Applied Sciences.

1.      Key words are four which need to be at least 5.

2.      The abstract is too short and need be enhanced for better understanding of what the authors have done.

3.      Any of the abbreviation should be written in full for the first time i.e the authors have used AIQS-DB directly without using the full wording i.e Application of Automated Identification and Quantification System with a Database.

4.      The references are random and need to be in serial i.e the authors have stated from reference no 26 instead of 1. I found this issue throughout the manuscript.

5.      In table 1 there is a column “compound number” The authors need to put an explanatory note for this i.e where this compound can be found using which library as all the author may not have access to the library. And to further facilitated the readers and gain citations my suggestion is to add one more column and write name of that compound of which number is given, this will make it more attractive and comprehensive.  

6.      The authors need to write subtopics to explain better the applications of AIQS-DB in various research fields i.e food, medicine, identification of the geographic origin, amount of sample needed, etc ets.   

7.      Conclusion need to be rephrased for more elaboration which will help in better understanding.

Author Response

The response is in attach file.

Reviewer 4 Report

What's the purpose of this research, I could not understand, that manuscript have been written poorly. Scheme of this manuscript seems to have not to been fictionalized. All of the GC/MS instruments have their respective m/s library (WILEY-NIST). Also, some firms have software packages for spectral libraries (e.g. Bio-rad, perkin-elmer). 

The instrument conditions for AIQS-DB seems limited to only two different devices, and  the authors did not mention any software package or application for this database.

The article should be rejected, with all due respect. This is my objective opinion.

The quality of English writing in this communication seems poor, it should be improved.

Round 2

Reviewer 1 Report

The authors made the suggested adjustments

Reviewer 4 Report

The abstract section is different from the review report form.  the authors continuously mention a database. But there is no database exists, and no links or additional data shared. Besides, there is no calibration curve for example data.  

Who is Professor Kadodami Kiwao, there is no info regarding himself on the internet.

Actually, I can not see any differences from the former version. This is not ethical, if you mention a database, you need to present a database (web-based, via GitHub, or GitLab, or directly its GUI via website) to the readers. Despite our previous recommendations, there has been no significant change exists.

Otherwise, the manuscript should not be accepted.

quality of english not too bad
